# Vaccination with Formulation of Nanoparticles Loaded with *Leishmania amazonensis* Antigens Confers Protection against Experimental Visceral Leishmaniasis in Hamster

**DOI:** 10.3390/vaccines11010111

**Published:** 2023-01-02

**Authors:** Marco Antonio Cabrera González, Ana Alice Maia Gonçalves, Jennifer Ottino, Jaqueline Costa Leite, Lucilene Aparecida Resende, Otoni Alves Melo-Júnior, Patrícia Silveira, Mariana Santos Cardoso, Ricardo Toshio Fujiwara, Lilian Lacerda Bueno, Renato Lima Santos, Tatiane Furtado de Carvalho, Giani Martins Garcia, Paulo Ricardo de Oliveira Paes, Alexsandro Sobreira Galdino, Miguel Angel Chávez-Fumagalli, Marília Martins Melo, Denise Silveira-Lemos, Olindo Assis Martins-Filho, Walderez Ornelas Dutra, Vanessa Carla Furtado Mosqueira, Rodolfo Cordeiro Giunchetti

**Affiliations:** 1Departamento de Morfologia, Universidade Federal de Minas Gerais (UFMG), Belo Horizonte 31270-901, MG, Brazil; 2Laboratório de Desenvolvimento Galênico e Nanotecnologia, Escola de Farmácia, Universidade Federal de Ouro Preto (UFOP), Ouro Preto 35400-000, MG, Brazil; 3Estación Experimental Agraria Baños del Inca, Instituto Nacional de Innovación Agraria, Cajamarca 06000, Peru; 4Departamento de Parasitologia, Universidade Federal de Minas Gerais (UFMG), Belo Horizonte 31270-901, MG, Brazil; 5Escola de Veterinária, Universidade Federal de Minas Gerais (UFMG), Belo Horizonte 31270-901, MG, Brazil; 6Laboratório de Biotecnologia de Microrganismos, Universidade Federal de São João Del-Rei (UFSJ), Campus Centro Oeste, Divinópolis 35501-296, MG, Brazil; 7Computational Biology and Chemistry Research Group, Vicerrectorado de Investigación, Universidad Católica de Santa María, Urb. San José S/N, Arequipa 04000, Peru; 8FIOCRUZ-Minas Gerais, Laboratório de Biomarcadores de Diagnóstico e Monitoração, Instituto René Rachou, Belo Horizonte 30190-002, MG, Brazil; 9Instituto Nacional de Ciência e Tecnologia em Doenças Tropicais—INCT-DT, Belo Horizonte 31270-901, MG, Brazil

**Keywords:** visceral leishmaniasis, nanotechnology, vaccine

## Abstract

Visceral leishmaniasis (VL) is a fatal disease caused by the protozoa *Leishmania infantum* for which dogs are the main reservoirs. A vaccine against canine visceral leishmaniasis (CVL) could be an important tool in the control of human and CVL by reducing the infection pressure of *L. infantum*. Despite the CVL vaccine available on the market, the Brazilian Ministry of Health did not implement the use of it in their control programs. In this sense, there is an urgent need to develop more efficient vaccines. In this study, the association between two polymeric nanoformulations, (poly (D, L-lactic) acid (PLA) polymer) loading *Leishmania amazonensis* antigens, was evaluated as a potential immunobiological agent against VL using golden hamsters as an experimental model. The results indicated that no significant adverse reactions were observed in animals vaccinated with LAPSmP. LAPSmP presented similar levels of total anti-*Leishmania* IgG as compared to LAPSmG. The LAPSmP and LAPSmG groups showed an intense reduction in liver and spleen parasitic load by qPCR. The LAPSmP and LAPSmG vaccines showed exceptional results, indicating that they may be promising candidates as a VL vaccine.

## 1. Introduction

Leishmaniasis comprises a group of diseases caused by different flagellated protozoa belonging to the genus *Leishmania* [1,2]. The most severe form of the disease, visceral leishmaniasis (VL), is caused by parasites of the *Leishmania donovani* complex, essentially by *L. donovani* and *L. infantum* (synonym *Leishmania chagasi* in South America). The parasites are transmitted to vertebrate hosts during the blood meal of blood-sucking sandflies females of the genera *Phlebotomus* or *Lutzomyia* [3,4,5]. In Latin America, Brazil is responsible for about 97% of reported cases [6] and a territorial expansion of VL has been reported in recent years [7].

Dogs represent the most important domestic reservoirs of *L. infantum* and vaccination of dogs is considered an important tool in the control of both human (HVL) and canine (CVL) visceral leishmaniasis cases. Controlling CVL is considered a fundamental measure to reduce the infection pressure of *L. infantum* [8]. Due to such importance, researchers have been devoting time and effort in the search for an anti-CVL vaccine and, to date, there are three commercial vaccines: CaniLeish^®^ and LetiFend^®^, available in Europe, and Leish-Tec^®^, marketed in Brazil. However, the Brazilian Ministry of Health has not yet incorporated the commercially available vaccine into its control programs. Therefore, it is still necessary to develop a vaccine with better efficacy to be available in health control programs against the disease. In this sense, different vaccine candidates against CVL have been reported to show the ability to induce immunoprotective mechanisms and a reduced parasite load [9,10,11,12,13,14,15,16,17,18,19].

One of the reasons that may explain the fact that vaccines do not reach a satisfactory level of protection is the lack of an adequate administration system [20]. Therefore, there is still a crucial need for an improved, safe, and efficient delivery system to enhance the immunogenicity of available vaccine candidates [21]. In order to circumvent this problem, biodegradable polymeric nanocarriers had emerged as a safe and powerful alternative to be employed in the control of various diseases associated with drugs or vaccines. Polymer-based nanoparticles are proficient in boosting the quality and magnitude of immune responses in a predictable fashion [22,23,24,25]. The use of nanoparticles as an antigen delivery system for VL has shown promising results, observed in studies of activation of the protective cellular immune response and parasite load reduction [26,27,28].

In recent years, our research group focused efforts on developing and analyzing heterologous vaccine candidates against CVL using antigens of the *Leishmania* dermatotropic species, with promising results [12,13,14,15,16,17]. In these studies, *L. braziliensis* was used to compose the LBSap vaccine, showing high immunogenicity profile resulting in heterologous protection against *L. infantum* infection in dogs [10,12,13,14,15,16,17]. In view of the promising results of heterologous protection using *L. braziliensis*, we aimed to test whether *L. amazonensis* would also be able to trigger protection against *L. infantum* infection, since it is one of the most important species causing cutaneous leishmaniasis in Brazil, found throughout the country and responsible for diffuse cutaneous leishmaniasis [29,30]. In this sense, the present study sought to develop a new CVL vaccine using nanoparticles loaded with *L. amazonensis* antigens to favor the delivery of the antigen to the presenting cells and allow a protective immune response against VL. Thus, a pre-clinical vaccine trial was carried out on hamsters to analyze the immunogenicity of this nanoparticle formulation against *L. infantum* infection.

## 2. Materials and Methods

### 2.1. Animals and Experimental Groups

This study was approved by the Ethics Committee on the Use of Animals of the Federal University of Minas Gerais (Protocol number 385/2013). Eight experimental groups with eight healthy male gold hamsters (*Mesocricetus auratus*) each, ages between 6–8 weeks, were used to test a vaccination protocol. The animals were divided into the following groups: (i) Infected control group, administered only with sterile saline solution (C); (ii) saponin control group, administered with the adjuvant saponin; (iii) empty submicrometric small particle vaccinated group (PSmP); (iv) empty submicrometric large particle vaccinated group (PSmG); (v) *Leishmania amazonensis* crude antigen vaccinated group (LA); (vi) *Leishmania amazonensis* crude antigen associated with the saponin group (LASAP); (vii) submicrometric small particle plus *L. amazonensis* crude antigen vaccinated group (LAPSmP); and (viii) submicrometric large particle plus *L. amazonensis* crude antigen vaccinated group (LAPSmG). In addition, eight animals were used as a pre-immune (PI) group.

### 2.2. Vaccine Preparation

The crude antigen of *L. amazonensis* was prepared using a reference strain (MHOM/BR/75/M2904), grown in *Minimum Essential Medium* (α–MEM—Gibco BRL, New York, USA), supplemented with 10% bovine fetal serum and 2 mM penicillin under biochemical oxygen demand (BOD) at 23 °C for a maximum of ten passages. On the fifth day of the culture, 10^8^ stationary promastigotes were sonicated, and the crude antigen was aliquoted and stored at −80 °C [10].

The nanospheres were prepared to employ the interfacial deposition method of a preformed polymer, as described by Fessi et al. (1989) [31], at Laboratório de Desenvolvimento Galênico e Nanotecnologia da Escola de Farmácia, Universidade Federal de Ouro Preto, Brazil. The nanospheres were prepared by the association of a poly-D, L-lactic acid polymer (PLA) solution and chitosan, previously solubilized in 0.05 M acetic acid at concentrations of 50 mg/mL (large submicrometric particle—PSmG) and 5 mg/mL (small submicrometric particle—PSmP). For the production of LAPSmG and LAPSmP, 1 mL of the *L. amazonensis* antigen suspension, at a concentration of 1 mg/mL, was added to the nanospheres. The size of the nanospheres was characterized by the polydispersity index and zeta potential, as described by Garcia et al. (2015) [32] and determined by a photon correlation spectroscopy analyzer in a Nanosizer N5 Submicron particle analyzer (Beckman Coulter, Newark, NJ, USA). Each vaccine formulation was diluted with 0.9% sterile saline buffer during its preparation and the final volume was adjusted for 100 µL/dose.

### 2.3. Experimental Protocol

The animals used in the study received three doses of the formulations, according to their respective group by the subcutaneous route at 21-day intervals. An evaluation of the safety and toxicity of the formulations based on nanoparticles was performed by observing the inoculum site (presence of nodules, papules, or ulcerative lesions), signs such as local pain, alopecia, and bristling and by evaluating animals’ behavior (aggression or apathy) [10]. Fifty-one days after the end of the vaccine protocol, the animals were challenged with 10^7^ *L. infantum* stationary promastigotes (strain C46) by the intraperitoneal route. The parasites were grown under the same conditions described by Giunchetti et al. (2007) [10]. At 180 days post-challenge (T180), a new blood sampling was performed 180 days post-challenge (T180) after which the animals were euthanized. The liver and spleen were collected to evaluate the vaccine efficacy, as well as make a comparative weight analysis between groups.

### 2.4. Assessment of Anti-Leishmania Total IgG Reactivity

The anti-*Leishmania* humoral immune response was evaluated by employing the sera obtained in PI, T30, and T180. Total IgG antibody was measured using an in-house*e* ELISA assay, as previously described [33]. Briefly, the plates were sensitized with 1 μg/mL per well with *L*. *infantum* crude antigen. Next, 200 μL/well of PBS-BSA solution (3% BSA) was added to prevent unspecific binding. The hamster’s serum diluted 1:100, was added and after incubation, the secondary anti-hamster IgG, diluted 1:5000, was incubated with the sample. The reaction was revealed with solution containing 0.03% hydrogen peroxide and 0.022 µg of ortho-phenylenediamine (OPD). The spectrophotometer ELX800 (Biotek Instruments VT, Winooski, VT, USA) was used to measure the wells’ optical density (OD) in a 490 nm wavelength.

### 2.5. Parasite Burden Evaluation by qPCR

The liver and spleen were used to evaluate the vaccine efficacy by qPCR. Previously to qPCR reaction, gDNA extraction from liver and spleen was performed by using a NucleoSpin^®^ Tissue extraction kit—Genomic DNA from tissue protocol (Macherey-Nagel, Duren, Germany) in accordance with the manufacturer’s instructions. DNA was resuspended in 50 μL of elution buffer (kit provided) and quantified using Nanodrop N2000 equipment (ThermoFisher Scientific, Waltham, MA, USA).

The qPCR reactions were performed using 5 μL of the 2× Power Syber^®^-Green ABI reagent (ThermoFisher Scientific, Waltham, MA, USA), 0.5 μL of each primer (2μM), (kDNA Forward- 5′ CGTGGGGGAGGGGCGTTCT 3′ and kDNA Reverse- 5′ CCGAAGCAGCCGCCCCTATT 3′) with a 135 pb amplicon size, and 4 μL of DNA (5 ng/μL), in a final volume of 10 μL per well. For normalization of the reaction, the β-actin constitutive hamster gene was selected using primers that amplify a 120 bp fragment (Forward- 5′ TCGTACGTGGGTGACGAGGC 3′ and reverse- 5′ GTAGAAGGTGTGGTGCCAGA 3′).

The amplification process consisted of pre-denaturation at 95 °C for 10 min, followed by 40 cycles of denaturation at 95 °C for 15 s, and annealing at 60 °C for 1 min. A melting curve was also used. The qPCR results were plotted using the standard curve method, which was assembled from the serial dilution of DNA extracted from 10^8^ *Leishmania* sp. promastigotes and 10^6^ hamster splenocytes. The spleen qPCR data was also used to analyze the correlation between parasite load and anti-*Leishmania* antibody.

### 2.6. Statistical Analyses

All statistical analyses were performed using GraphPad Prism 5.0 software (Prism Software, California, CA, USA). Data normality was demonstrated by the Kolmogorov–Smirnoff test. The analysis of variance (ANOVA) tests, followed by the Tukey multiple comparisons test, were used to compare between the groups. The Kruskal–Wallis test, followed by Dunn’s test, was used for the nonparametric qPCR data. Statistical differences were considered significant when the *p*-value was less than 0.05. Spearman’s correlation methods were applied to investigate the correlation between splenic parasite load and antibody production.

## 3. Results

### 3.1. Characterization of Submicrometric Particle

The mass of PLA and chitosan was varied in order to produces sub-micrometric particles with different sizes and capable of encapsulating the vaccine antigen (Table 1). The small submicrometric particle (PSmP) sized 350.6 nm and the zeta potential was +66.6 mV. The large submicrometric particle (PsmG) sized 613.4 nm and the zeta potential was +66.8 mV. The antigen encapsulation does not alter significantly the overall size of the submicrometric particles. LAPSmP sized 409.8 nm (zeta potential was +42.2 mV) and LAPSmG sized 652.7 nm (zeta potential was +56.3 mV) (Table 1).

### 3.2. Safety Analyses

The animals were observed after each inoculum for possible adverse post-vaccination reactions. The presence of non-ulcerated nodules, of firm consistency, were observed in two animals from the PSmG group. Moreover, one animal from the LAPSmG group developed an ulcerated nodule that measured 3 × 3 mm. Although these nodules were transiently observed, the animals showed no signs of pain or behavioral changes. After 180 days of *L. infantum* experimental infection, the control groups (C, PSmG, PSmG, and LA) were those with more evident dermatological changes, such as alopecia, as compared to animals vaccinated with LAPSmG and LAPSmP, as shown in Figure 1.

### 3.3. Serum Reactivity of Anti-Leishmania Total IgG

Thirty days after the last inoculum, the PSmP, LA, LASAP, LAPSmP, and LAPSmG groups exhibited higher levels of total anti-*Leishmania* IgG antibodies in relation to PI (*p* < 0.05; *p* < 0.001; *p* < 0.0001; *p* < 0.01; *p* < 0.001, respectively—Figure 2A). The LA, LASAP, LAPSmP, and LAPSmG groups showed increased total anti-*Leishmania* IgG production, as compared with C (*p* < 0.05; *p* < 0.0001; *p* < 0.05; *p* < 0.01, respectively—Figure 2A). LASAP and LAPSmG displayed high IgG levels as compared to SAP group (*p* < 0,.01; *p* < 0.05, respectively—Figure 2A). Furthermore, LASAP produced higher IgG levels as compared to the LAPSmP (*p* < 0.01) and LA groups (*p* < 0.01) (Figure 2A). The LAPSmG group showed an increased in IgG production relative to the PSmG group (*p* < 0.01). After the experimental challenge (Figure 2B), the C, SAP, PSmP, PSmG, LA, LASAP, and LAPSmG groups presented higher total IgG levels as compared with PI (*p* < 0.01; *p* < 0.05; *p* < 0.01; *p* < 0.05; *p* < 0.01; *p* < 0.05; *p* < 0.001, respectively). Moreover, the LAPSmG group showed an increase in IgG levels as compared to the control group (*p* < 0.05).

### 3.4. Vaccine Efficacy Evaluation

The weights of the spleen and liver were compared after the animals had been euthanized. No statistical differences were observed in the liver weight (Figure 3A,C). In contrast, reduced spleen weights were observed in the SAP group relative to the LA and LASAP group (*p* < 0.05 for both). Moreover, the LAPSmG group presented reduced spleen weights in relation to the LASAP group (*p* < 0.05) (Figure 3B,C).

Data from the qPCR analysis showed that LAPSmP and LAPSmG were the only groups with a marked reduction in the splenic parasite load (*p* < 0.01; *p* < 0.05, respectively) (Figure 4A) in relation to the control group. Similar results were observed in the liver parasite load for the LAPSmP and LAPSmG groups displaying an intense reduction in relation to the control group (*p* < 0.05 for both). Furthermore, PSmP group also showed reduction in liver parasite load in relation to the control group (*p* < 0.05) (Figure 4B).

The correlation between splenic parasite load and antibody production was analyzed (Figure 5). The results showed a positive correlation between the parasite load and the IgG reactivity in the control (*p* < 0.011; r = 0.488) and SAP groups (*p* < 0.006; r = 0.876) (Figure 5). No correlation was observed between these variables in the other groups.

## 4. Discussion

CVL represents a major veterinary and public health problem in various parts of the New and Old World [1]. In Brazil, CVL autochthonous cases have been reported in several regions that had no previous record of canine cases [34,35]. Currently, 25 (25/27) Brazilian states have reported autochthonous CVL cases, highlighting Brazil as the principal country in South America reporting this neglected disease [36]. Thus, due to an increase in the incidence of VL cases, the development of new strategies, such as prophylactic vaccines to prevent the infection in dogs, has become a top priority [6]. In fact, dog vaccination is the only viable way to control this zoonotic disease [37]. Despite advances, the Brazilian Ministry of Health has not introduced the only commercially available vaccine, Leish-Tec^®^, into its VL control program. To date, this vaccine is recommended only as individual prophylaxis in dogs [38]. The development of effective vaccines is still required to induce an effective form of protection against *L. infantum* infection in dogs with helpful results to VL health policies.

The development of an ideal vaccine for CVL needs to meet certain criteria: It must be safe, reproducible, allow large-scale production, and be cost-effective [39]. Furthermore, one of the most critical points, if not the most essential, is the activation of a lasting immune response against *L. infantum* infection [10,11,12,13,14,15,16,17]. An ideal vaccine against VL requires triggering a prominent type 1 immune response. Additionally, it is important to stress the importance of selecting an appropriate antigen, the right adjuvant, and/or delivery vehicle [40,41]. Nanoparticulate formulations are currently considered as ideal vaccine delivery systems [22,23,24,40]. In fact, substantial progress has been made in the area of nanotechnology for vaccines [42] and, to date, there are studies on such diseases as malaria [43], hepatitis B virus [44], HIV [45], H1N1 [46], West Nile virus [47], and *Schistosoma mansoni* [48]. Thus, polymeric nanoparticles are an attractive alternative to be used in developing effective VL vaccines, acting as carriers of *Leishmania* antigens due to their improved bioavailability and lower toxicity [40,49].

In this study, the association between two polymeric nanoformulations with *L. amazonensis* antigens was analyzed to evaluate their potential as an immunobiological agent against VL using golden hamsters as experimental model. This animal model was chosen because of its greater similarity to the course of VL in dogs and humans [33,50]. However, the lack of immunological reagents for this animal model makes it difficult to understand the immunological mechanisms induced by vaccination. In this sense, the limitation to evaluate the cellular immune response in addition to the production of cytokines are the main negative points of the hamster’s animal model in pre-clinical vaccine trials. Furthermore, the VL ongoing was marked by a high production of anti-*Leishmania* antibodies and a positive correlation between antibody production and disease progression has already been reported [33,51,52]. The evaluation of parasite load is the main parameter to be evaluated for the analysis of efficacy, regardless of the animal model used in the pre-clinical test. Despite the limitation of evaluation in the cellular immune response in hamsters, this animal model is highly susceptible to *L. infantum* infection, which is a critical factor for the analysis of vaccine efficacy.

The evaluation of the innocuity and toxicity of tested immunobiological agents should not induce local or systemic damage that is incompatible with their administration [53]. An analysis of clinical and laboratory parameters was carried out to verify the safety and toxicity of the proposed vaccines. No local adverse reactions were observed in inoculated animals with PSmP and LAPSmP. In contrast, the hamsters immunized with PSmG and LAPSmG exhibited transitory local reactions. However, vaccination using LAPSmP and LAPSmG was not associated with behavioral or clinical changes, lymphadenopathy, or any other general adverse reactions. Moreover, no animal weight changes or clinical or laboratory exam abnormalities were observed (data not shown). Thus, the overall tolerance of the candidate nanoparticles vaccines in hamsters appeared to be adequate. The size of the nanoparticle used for immunization influences the cell specificity as well as the way it will be transported to the draining lymph nodes [21]. This could be the reason why local reactions were observed only in the groups immunized with the large submicrometric particle. A recent study published by our research group, also using nanoparticles loaded with *Leishmania braziliensis* antigens to immunize hamster, did not result in adverse reactions in the vaccinated large submicrometric particle group. However, the size of the *Leishmania* antigen loaded large submicrometric particle was smaller (570 nm) than the particle size used in the present study (653 nm) [54]. After the *L*. *infantum* experimental challenge, dermatological changes, mainly alopecia, were observed, especially in the Control, SAP, PSmP, PSmG, LA, and LASAP groups. These changes are consistent with those already reported during the course of VL infection in hamsters [50,55].

The levels of anti-*Leishmania* total IgG were accessed for antigenicity analysis and thirty days following the last dose, the LA, LASAP, LPSmP, and LPSmG groups showed an increase in the production of antibodies. In agreement with these findings, other heterologous antigens used against VL infection showed a high anti-*Leishmania* production after vaccination [13,14,56]. These data demonstrate that the vaccines activated a strong antigenicity profile. In addition, after the experimental challenge, the levels of total IgG in those groups increased and all the infected groups showed a similar pattern of IgG production. Mendonça and colleagues (2016) observed that the LBSap, a vaccine containing *Leishmania braziliensis* crude antigen, induced higher levels of anti-*Leishmania* IgG, as well as a reduction in spleen *L*. *infantum* parasite load in infected and vaccinated animals [57]. Similarly, Ottino et al. (2022) observed elevated levels of the anti-*Leishmania* antibody with a reduced spleen parasite load in infected and vaccinated hamsters [54].

Notably, hepatomegaly and splenomegaly are among the most evident clinical findings in VL, having been reported in studies with dogs and murine models [58,59,60]. In the present study, no significant changes in liver and spleen weight were observed in the LAPSmP and LAPSmG vaccinated groups in relation to the control group. Moreira et al. (2016) analyzed different clinical changes in distinct *L. infantum* strains and inoculation routes in hamsters and demonstrated changes in liver weight only after several months of experimental infection [50]. In this sense, the authors of this study have hypothesized that the absence of a difference in the weight of the organs between the vaccinated and the control groups may be due to the time of infection when euthanasia was performed (180 days post-challenge). Probably, the time required to trigger more prominent inflammatory changes in *L. infantum* target organs after experimental challenge was short, as previously described [50].

The vaccine efficacy was evaluated by qPCR to quantify the parasite burden in the liver and spleen. This quantification makes it possible to evaluate the potential of vaccine candidates to control the *L. infantum* infection [55]. The parasitological data revealed that only the LAPSmP and LAPSmG groups were able to reduce the parasite load in both liver and spleen, demonstrating a notable capacity in controlling parasite spread in hamsters. Similarly, corroborating the potential of the application of nanoparticles for vaccines against VL, other studies also reported a reduction in the *L*. *infantum* parasite load in infected and vaccinated animals [26,27,28,54,61]. In agreement with these findings, no correlation was observed between the parasite load and antibody production in these vaccinated groups, despite the elevated levels of IgG produced by the LAPSmP and LAPSmG groups. In fact, a correlation between splenic parasite load and anti-*Leishmania* antibody was detected only in the control and SAP groups and several studies have already demonstrated a direct relationship between tissue parasite density, clinical status and antibody titers in CVL [62,63,64,65]. Taken together, this study’s data confirmed the hypothesis that the development of vaccines using nanoparticles is a promising tool for VL vaccines and the LAPSmP and LAPSmG vaccine formulations hold promise for future vaccination strategies against CVL.

## 5. Conclusions

The vaccine formulations proposed in this study, LAPSmP and LAPSmG, proved to be innocuous, aside from transient local reactions observed in the LAPSmG group, since no safety concerns were identified, plus they proved to be antigenic. Although the immunological mechanisms activated by the immunization have not yet been investigated, it can be hypothesized that they activated a protective cellular response since a drastic reduction in the liver and spleen parasite load was observed in both LAPSmP and LAPSmG vaccinated groups. In this sense, the present findings encourage the use of nanoparticles in association with *Leishmania* antigens for the development of new effective vaccines for VL control.

## Figures and Tables

**Figure 1 vaccines-11-00111-f001:**
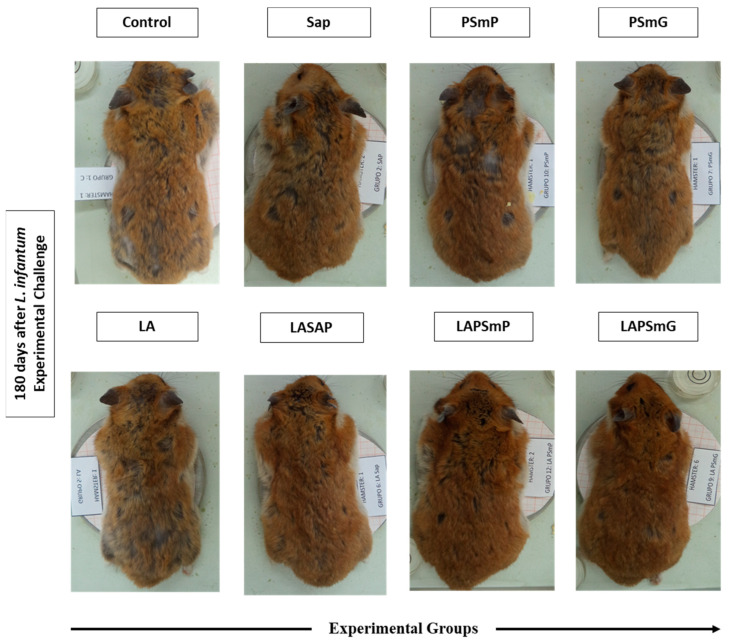
Macroscopic changes observed in the dorsal region of golden hamsters up to l the date of euthanasia, in the different experimental groups, including: C—control; SAP—saponin; PSmP—small empty submicron particle; PSmG—large empty submicron particle; LA—*L*. *amazonensis* crude antigen; LASAP—*L*. *amazonensis* crude antigen associated with saponin; LAPSmP—small submicron particle associated with *L*. *amazonensis*; LAPSmG—large submicron particle associated with *L*. *amazonensis*.

**Figure 2 vaccines-11-00111-f002:**
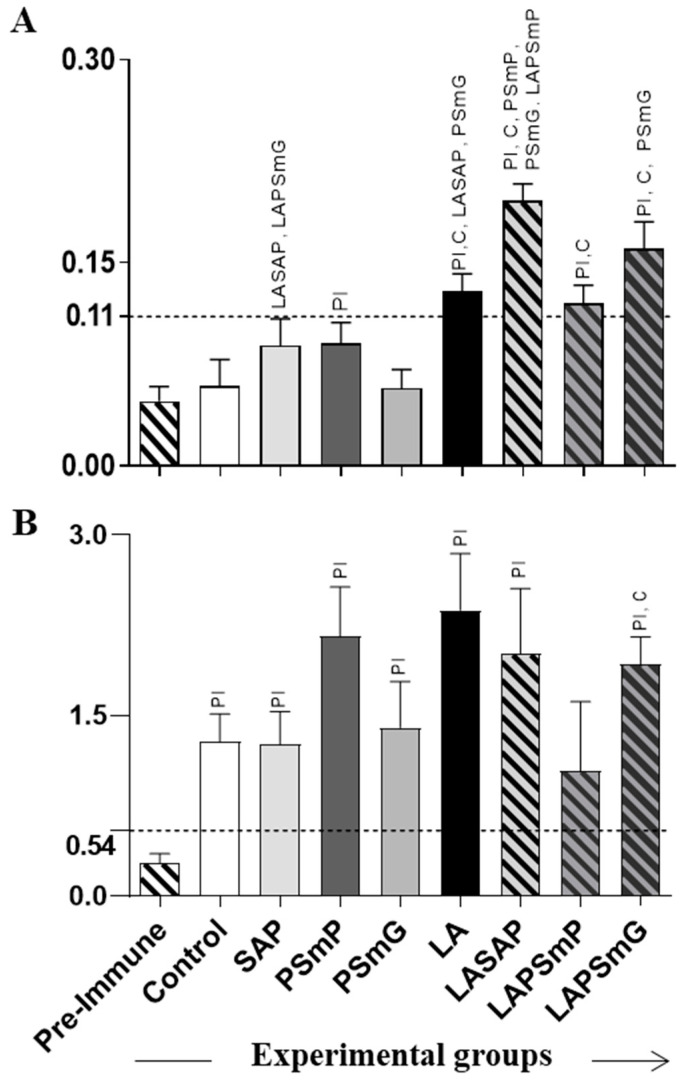
Anti-*Leishmania infantum* total IgG production. Hamster serum was obtained (**A**) 30 days after the third dose and (**B**) 180 days after the experimental challenge. The graphs represent the mean values and standard deviation of total IgG optical density measured by the ELISA assay for each group. The cut-off illustrated as a dotted line in the graph and used to assess the reactivity present in the samples is represented by the average of the absorbance values of the pre-immune serum sample added to the value of twice the standard deviation. The statistical difference (*p* < 0.05) was represented by acronyms of the corresponding group. Legend: PI: pre-immune; C: control; SAP: saponin; PSmP: small empty submicron particle; PSmG: large empty submicron Particle; LA: *L*. *amazonensis* crude antigen; LASAP: *L*. *amazonensis* crude antigen associated with saponin; LAPSmP: small submicron particle associated with *L*. *amazonensis*; LAPSmG: large submicron particle associated with *L*. *amazonensis*.

**Figure 3 vaccines-11-00111-f003:**
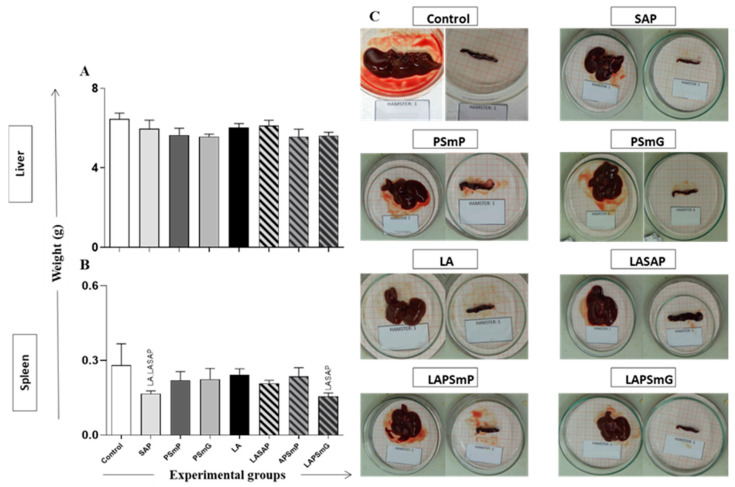
Liver (**A**) and spleen (**B**) weights were measured immediately after euthanasia (T180) and are represented as mean ± standard deviation. (**C**) Contains the representative image of the organs (liver—left side and spleen—right side) of each group. The statistical difference (*p* < 0.05) was represented by the acronyms of the corresponding group. Legend: C: control; SAP: saponin; PSmP: small empty submicron particle; PSmG: large empty submicron particle; LA: *L*. *amazonensis* crude antigen; LASAP: *L*. *amazonensis* crude antigen associated with saponin; LAPSmP: small submicron particle associated with *L*. *amazonensis*; LAPSmG: large submicron particle associated with *L*. *amazonensis*.

**Figure 4 vaccines-11-00111-f004:**
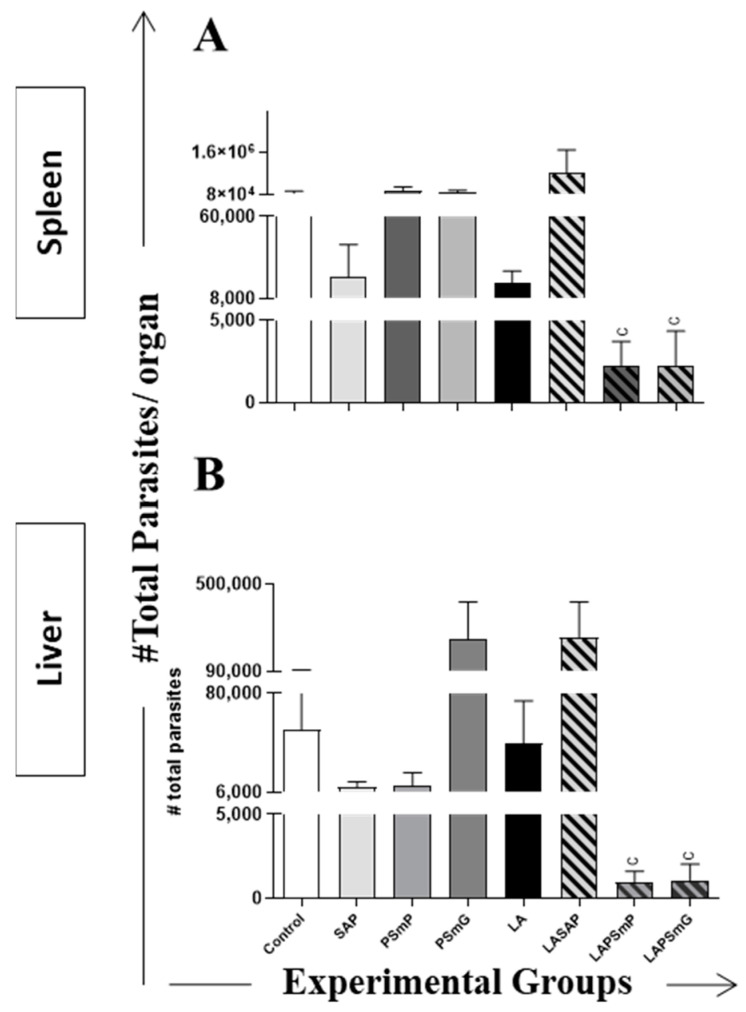
Parasite load in the spleen (**A**) and liver (**B**) after *L*. *infantum* experimental challenge, according to qPCR reaction. The x axis represents the experimental groups and the y axis represents the mean ± standard deviation of parasite quantification in 10^6^ cells of each organ. The statistical difference (*p* < 0.05), when present, was represented by the acronym of the corresponding group. Legend: C: control; SAP: saponin; PSmP: small empty submicron particle; PSmG: large empty submicron particle; LA: *L*. *amazonensis* crude antigen; LASAP: *L*. *amazonensis* crude antigen associated with saponin; LAPSmP: small submicron particle associated with *L*. *amazonensis*; LAPSmG: large submicron particle associated with *L*. *amazonensis*.

**Figure 5 vaccines-11-00111-f005:**
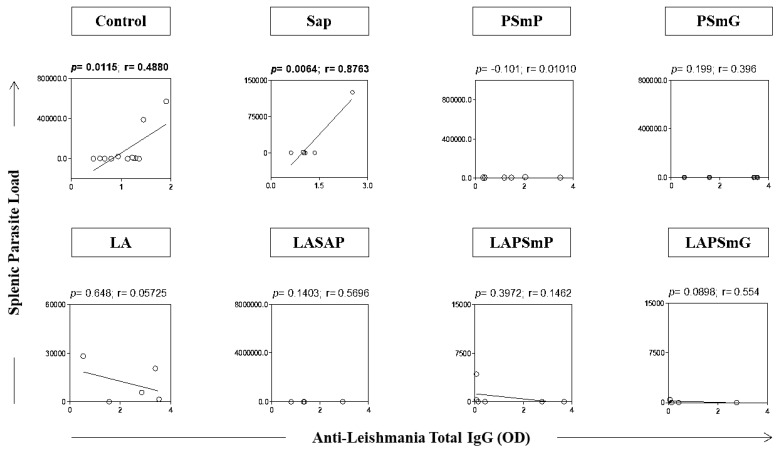
Correlation between splenic parasite load and total anti-*Leishmania* total IgG levels displayed as “◯”. The statistical difference was considered when *p* < 0.05. Legend: SAP: saponin; PSmP: small empty submicron particle; PSmG: large empty submicron particle; LA: *L. amazonensis* crude antigen; LASAP: *L. amazonensis* crude antigen associated with saponin; LAPSmP: small submicron particle associated with *L. amazonensis*; LAPSmG: large submicron particle associated with *L. amazonensis*.

**Table 1 vaccines-11-00111-t001:** Characteristics of the SMP preparations.

	PLA (mg/mL)	Chitosan (mg/mL)	Antigen (1 mg/mL)	Size (nm)	Polydispersity	Zeta Potential (mV)
PSmP	5	5	-	350.6 ± 15.3	0.159	+66.6 ± 4.3
PSmG	50	50	-	613.4 ± 16.8	0.133	+66.8 ± 4.4
LAPSmP	5	5	*L. amazonensis*	409.8 ± 11.7	0.157	+42.2 ± 5.8
LAPSmG	50	50	*L. amazonensis*	652.7 ± 12.3	0.153	+56.3 ± 4.3

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
