# Peer review of "Vaccination with Formulation of Nanoparticles Loaded with Leishmania amazonensis Antigens Confers Protection against Experimental Visceral Leishmaniasis in Hamster"

_vaccines, 2023, doi:10.3390/vaccines11010111_

Round 1

Reviewer 1 Report

In this manuscript, González et al. evaluated the safety and protection of vaccines using two polymeric nanoparticle formulations incorporating Leishmania antigens in golden hamster model. The authors analyzed the adverse reactions, antigen-binding IgG responses, and animal protection profiling based on qPCR of parasitic load in spleen and liver. Though some reduction of parasite load has been observed in LAPSmP and LAPSmG groups, they did not correlate with antibody production. The experiments for this study are incomplete, which led to weak conclusions.

The manuscript lacks enough explanation about the development of the vaccine candidate in this study, including heterologous vaccine selection, the reasons why the authors chose L. amazonensis as vaccine candidate to induce immune response against L. infantum, as well as further explanations about the two different nanoparticle formulations - submicrometric small and large particles. Also the authors didn’t study the purity and other characteristics of the vaccine content itself.

For data presentation, the figures didn’t clearly show the results in a more quantitative way, especially in Figure 1. In addition, there was no significant change induced after virus challenge even in the control group, therefore it is difficult to investigate the difference in the protective effect by vaccination. The authors only used one parameter (parasite load in spleen/liver) to evaluate the infection level, it may be necessary to establish a better infection model for this study with more parameters.

For vaccine candidates against parasites like Leishmania, one of their most important features is whether they could effectively induce CD4+ and CD8+ T cell responses. However, in this manuscript, the authors only studied the antibody responses, limiting to their binding levels against crude antigens, while no analysis of T cell responses has been performed at all. This may also be the reason why there was no correlation observed between splenic parasite load and antibody production as shown in Figure 5. Therefore, the conclusion of the analysis may be too weak.

Author Response

Review Report 1

In this manuscript, González et al. evaluated the safety and protection of vaccines using two polymeric nanoparticle formulations incorporating Leishmania antigens in golden hamster model. The authors analyzed the adverse reactions, antigen-binding IgG responses, and animal protection profiling based on qPCR of parasitic load in spleen and liver. Though some reduction of parasite load has been observed in LAPSmP and LAPSmG groups, they did not correlate with antibody production. The experiments for this study are incomplete, which led to weak conclusions.

We acknowledge the Reviwere#1 for this comment. In fact, the antibody production is an important feature in visceral leishmaniasis progression (REIS et al., 2006) and the uncontrolled concentration of antibodies and the presence of Leishmania antigens induce the formation of circulating immune complexes. These immune complexes are responsible for some of the clinical manifestations, including renal failure, the main clinical manifestation, which leads to the death of the infected animal (CACHEIRO‐LLAGUNO et al., 2021). As the hamster model presents great similarity to the progression of the disease in humans and dogs, elevated antibody production is also detected in this animal model as biomarker for disease ongoing (Moreira et al., 2012). Concomitantly, clinical alterations are detected in these animals (Moreira et al., 2012). Besides the antibody production cannot be associated to protection against visceral leishmaniasis, it is considered an important feature to indicate antigenicity in vaccine evaluation. Corroborating with our methodology, the study conducted by Taslimi et al. (2019) also evaluated only the production of antibodies and the parasitic load of vaccinated and infected hamsters. However, the lack of specific reagents for hamsters, that could allow evaluating the parameters of innate and acquired cellular immunity activated by vaccination is the main negative point of this animal model (Loría-Cervera et al., 2014). In contrast, as this animal model is highly susceptible to L. infantum infection, the analysis of vaccine candidates is interesting, especially when a reduction in the parasite load is observed, which is the main biomarker used to evaluate the vaccine efficacy in hamster model.

CACHEIRO‐LLAGUNO, C. et al. Role of circulating immune complexes in the pathogenesis of canine leishmaniasis: New players in vaccine development. Microorganisms, v. 9, n. 4, p. 712, 2021.

Loría-Cervera EN, Andrade-Narváez FJ. Animal models for the study of leishmaniasis immunology. Rev Inst Med Trop Sao Paulo. 2014 Jan-Feb;56(1):1-11. doi: 10.1590/S0036-46652014000100001.

Moreira, D.; Vitoriano-souza, J.; Roatt, B. M.; Melo, P.; Vieira, D. A.; Ker, H. G.; Mirelle, J.; Cardoso, D. O.; Giunchetti, R. C.; Carneiro, M.; Lana, M.; Reis, A. B. Parasite Burden in Hamsters Infected with Two Different Strains of Leishmania (Leishmania) infantum: “Leishman Donovan Units” versus Real-Time PCR. PLoS One 2012, 7, 10, e47907, doi: 10.1371/journal.pone.0047907.

REIS, A. B. et al. Phenotypic features of circulating leucocytes as immunological markers for clinical status and bone marrow parasite density in dogs naturally infected by Leishmania chagasi. Clinical and Experimental Immunology, v. 146, n. 2, p. 303–311, 2006.

Taslimi, Y., Zahedifard, F., Taheri, T., Doroud, D., Dizaji, S.L., Saljoughian, N., Rafati, S., 2020. Comparison of Protective Potency of DNA and Live Vaccines Expressing A2-CPA-CPB 15, 383–392. doi.org/10.18502/ijpa.v15i3.4203

The manuscript lacks enough explanation about the development of the vaccine candidate in this study, including heterologous vaccine selection, the reasons why the authors chose L. amazonensis as vaccine candidate to induce immune response against L. infantum, as well as further explanations about the two different nanoparticle formulations - submicrometric small and large particles. Also the authors didn’t study the purity and other characteristics of the vaccine content itself.

Our research group has been working with the development of heterologous vaccines, obtaining promising results using L. braziliensis antigens for protection against L. infantum (Aguiar-Soares et al., 2014; 2020; Giunchetti et al., 2007; 2008a; 2008b; Mendonça et al., 2016; Ottino et al., 2022; Resende et al., 2013; 2016; Roatt et al., 2012).

Aguiar-Soares, R. D.; Roatt, B. M.; Gama-Ker, H.; Moreira, N. D.; Mathias, F. A. S.; Cardoso, J. M. O.; et al. LbSapSal-vaccinated dogs exhibit increased circulating T-lymphocyte subsets (CD4+ and CD8+) as well as a reduction of parasitism after challenge with Leishmania infantum plus salivary gland of Lutzomyia longipalpis. Parasit Vectors 2014, 7, 61, doi:10.1186/1756-3305-7-61.

Aguiar-Soares, R. D. de O.; Roatt, B. M.; Mathias, F. A. S.; Reis, L. E. S.; Cardoso, J. M. de O.; Brito, R. C. F. de; Ker, H. G.; Corrêa-Oliveira, R.; Giunchetti, R. C.; Reis, A. B. Phase I and II Clinical Trial Comparing the LBSap, Leishmune®, and Leish-Tec® Vaccines against Canine Visceral Leishmaniasis. Vaccines 2020, 8, 690, doi:10.3390/vaccines8040690.

Giunchetti, R. C.; Correa-Oliveira, R.; Martins-Filho, O. A.; Teixeira-Carvalho, A.; Roatt, B. M.; Aguiar-Soares, R. D. O.; Souza, J. V.; Moreira, N. D.; Malaquias, L. C. C.; Castro, L. L. M.; Lana, M.; Reis, A. B. Immunogenicity of a killed Leishmania vaccine with saponin adjuvant in dogs. Vaccine 2007, 25, 7674–7686, doi: /10.1016/j.vaccine.2007.08.009.

Giunchetti, R. C.; Correa-Oliveira, R.; Martins-Filho, O. A.; Teixeira-Carvalho, A.; Roatt, B. M.; Aguiar-Soares, R.D. de O.; Coura-Vital, W.; Abreu, R. T.; Malaquias, L. C. C.; Gontijo, N. F.; Brodskyn, C.; Oliveira, C. I.; Costa, D. J.; Lana, M.; Reis, A. B. A killed Leishmania vaccine with sand fly saliva extract and saponin adjuvant displays immunogenicity in dogs. Vaccine 2008a, 265, 623-638, doi: 10.1016/j.vaccine.2007.11.057.

Giunchetti, R. C.; Reis, A. B.; Silveira-Lemos, D.; Martins-Filho, O. A.; Corrêa-Oliveira, R.; Bethony, J.; Vale, A. M.; Quetz, J. S.; Bueno, L. L.; França-Silva, J. C.; Nascimento, E.; Mayrink, W.; Fujiwara, R. T. al. Antigenicity of a whole parasite vaccine as promising candidate against canine leishmaniasis. Res VetSci 2008b, 85, 106–112, doi:10.1016/j.rvsc.2007.09.008.

Mendonça, L. Z.; Resende, L. A.; Lanna, M. F.; Aguiar-Soares, R. D. de O.; Roatt, B. M.; Castro, R. A. de O.; Batista, M. A.; Silveira-Lemos, D.; Gomes, J. de A. S.; Fujiwara, R. T.; Rezende, S. A.; Martins-Filho, O. A.; Corrêa-Oliveira, R.; Dutra, W. O.; Reis, A. B.; Giunchetti, R. C. Multicomponent LBSap vaccine displays immunological and parasitological profiles similar to those of Leish-Tec® and Leishmune® vaccines against visceral leishmaniasis. Parasites and Vectors 2016, 9, 1, doi: 10.1186/s13071-016-1752-6.

Ottino, J.; Leite, J. C.; Melo-Júnior, O. A.; González, M. A. C.; Carvalho, T. F. De.; Garcia, G. M.; Batista, M. A.; Silveira, P.; Cardoso, M. S.; Bueno, L. L.; Fujiwara, R. T.; Santos, R. L.; Paes, P. R. de O.; Silveira-Lemos, D.; Martins-Filho, O. A.; Galdino, A. S.; Chávez-Fumagalli, M. A.; Dutra, W. O.; Vanessa Carla Furtado Mosqueira, V. C. F.; Giunchetti, R. C. Nanoformulations with Leishmania braziliensis Antigens Triggered Controlled Parasite Burden in Vaccinated Golden Hamster (Mesocricetus auratus) against Visceral Leishmaniasis. Pathogens 2022, 10, 11, doi: 10.3390/vaccines10111848.

Resende, L. A.; Roatt, B. M.; Aguiar-Soares, R. D.; Viana, K. F.; Mendonça, L. Z.; Lanna, M. F.; Silveira-Lemos, D.; Cor-rêa-Oliveira, R.; Martins-Filho, O. A.; Fujiwara, R. T.; Carneiro, C. M.; Reis, A. B.; Giunchetti, R. C. Cytokine and nitric oxide patterns in dogs immunized with LBSap vaccine, before and after experimental challenge with Leishmania chagasi plus saliva of Lutzomyia longipalpis. Vet Par 2013, 198, 371–381, doi: 10.1016/j.vetpar.2013.09.011.

Resende, L. A.; Aguiar-Soares, R. D.; Gama-Ker, H.; et al. Impact of LbSapSal Vaccine in Canine Immunological and Parasitological Features before and after Leishmania chagasi-Challenge. PLoS One 2016, 11, e0161169, doi:10.1371/journal. pone.0161169

Roatt, B. M.; Aguiar-Soares, R. D.; Vitoriano-Souza, J.; Coura-Vital, W.; Braga, S. L.; Corrêa-Oliveira, R.; Martins-Filho, O. A.; Carvalho, A. T.; Lana, M.; Gontijo, N. F.; Marques, M. J.; Giunchetti, R. C.; Reis, A. B. Performance of LBSap vaccine after in-tradermal challenge with L. infantum and saliva of Lu. longipalpis: immunogenicity and parasitological evaluation. PLoS One 2012, 7, e49780, doi:10.1371/journal.pone.0049780.

In view of these results, we aimed to test whether the L. amazonensis specie could provide heterologous protection in vaccinated and L. infantum infected animals. In this sense, we justify the choice of L. amazonensis used as heterologous protection tested in the present study. We have included in the Introduction section the missing information, as follows:

 “(…) In recent years, our research group has focused efforts on developing and analyzing heterologous vaccine candidates against CVL using antigens of the Leishmania dermatotropic species, with promising results [12-17]. In these studies, L. braziliensis was used to compose the LBSap vaccine, showing high immunogenicity profile resulting in heterologous protection against L. infantum infection in dogs [10, 12-17]. In view of the promising results of heterologous protection using L. braziliensis, we aimed to test whether L. amazonensis would also able to trigger protection against L. infantum infection, (…)”

Moreover, we have added further information about the submicrometric particle preparation, as follows:

“(…) 3.1 Characterization of submicrometric particle

The mass of PLA and chitosan was varied in order to produces sub-micrometric particles with different sizes and capable of encapsulating the vaccine antigen (Table 1). The small submicrometric particle (PSmP) sized 350,6 nm and the zeta potential was +66,6 mV. The large submicrometric particle (PsmG) sized 613,4 nm and the zeta potential was +66,8 mV. The antigen encapsulation does not alter significantly the overall size of the submicrometric particles. LAPSmP sized 409,8 nm (zeta potential was +42,2 mV) and LAPSmG sized 652,7 nm (zeta potential was +56,3 mV) (Table 1). (…)”

For data presentation, the figures didn’t clearly show the results in a more quantitative way, especially in Figure 1. In addition, there was no significant change induced after virus challenge even in the control group, therefore it is difficult to investigate the difference in the protective effect by vaccination. The authors only used one parameter (parasite load in spleen/liver) to evaluate the infection level, it may be necessary to establish a better infection model for this study with more parameters.

Among the available experimental models, the hamster is an important animal model to study visceral leishmaniasis progression, as it is an experimental model that presents the greatest similarity with disease progression in both humans and dogs (Garg and Dube, 2006; Moreira et al., 2012; 2016). The main parameters used to assess disease progression as well as vaccine efficacy in this model are hematological and biochemical parameters; macroscopic changes; antibody production and parasite load (the last parameter is the most important during the evaluation of vaccine efficacy using this model). In this sense, we have described the clinical manifestations of L. infantum infection using representative pictures of each analyzed group (Figure 1) highlighting dermatological clinical changes in control groups when compared to vaccinated (LAPSmG and LAPSmP).

However, despite the great similarity of the disease progression of visceral leishmaniasis in human and dog, the hamster model lack of specific reagents for evaluating of cellular immune response triggered by vaccination (Loría-Cervera et al., 2014). Although there are several reagents to investigate the immune response for the experimental mice model, such model was not chosen due to dichotomy of Th1 and Th2 (induce a limitation of L. infantum infection ongoing) that is not observed in dogs and hamsters (Garg and Dube, 2006; Maia and Campino, 2018). In this sense, the reliable parameters to evaluate the vaccine efficacy in hamster model is the antibody production and parasite load. As it is known that visceral leishmaniasis progression is marked by a high production of anti-Leishmania antibodies (Reis et al., 2006), the positive correlation between the parasite load and antibody production detected only in Control and Sap groups (Fig. 5) reaffirms the vaccine ability to induce biomarkers related to resistance profile against the disease ongoing.

Garg R, Dube A. Animal models for vaccine studies for visceral leishmaniasis. Indian J Med Res. 2006 Mar;123(3):439-54.

Loría-Cervera EN, Andrade-Narváez FJ. Animal models for the study of leishmaniasis immunology. Rev Inst Med Trop Sao Paulo. 2014 Jan-Feb;56(1):1-11. doi: 10.1590/S0036-46652014000100001.

Maia C and Campino L (2018). Biomarkers Associated With Leishmania infantum Exposure, Infection, and Disease in Dogs. Front. Cell. Infect. Microbiol. 8:302. doi: 10.3389/fcimb.2018.00302

Moreira, D.; Vitoriano-souza, J.; Roatt, B. M.; Melo, P.; Vieira, D. A.; Ker, H. G.; Mirelle, J.; Cardoso, D. O.; Giunchetti, R. C.; Carneiro, M.; Lana, M.; Reis, A. B. Parasite Burden in Hamsters Infected with Two Different Strains of Leishmania (Leishmania) infantum: “Leishman Donovan Units” versus Real-Time PCR. PLoS One 2012, 7, 10, e47907, doi: 10.1371/journal.pone.0047907.

Moreira, D. N. Das; Vitoriano-Souza, J.; Roatt, B. M.; Vieira, P. M. de A.; Coura-Vital, W.; Cardoso, J. M. de O.; Rezende, M. T.; Ker, H. G.; Giunchetti, R. C.; Carneiro; C. M.; Reis, A. B. Clinical, hematological and biochemical alterations in hamster (Mesocricetus auratus) experimentally infected with Leishmania infantum through different routes of inoculation. Parasites and Vectors 2016, 9, 1, doi: 10.1186/s13071-016-1464-y.

REIS, A. B. et al. Phenotypic features of circulating leucocytes as immunological markers for clinical status and bone marrow parasite density in dogs naturally infected by Leishmania chagasi. Clinical and Experimental Immunology, v. 146, n. 2, p. 303–311, 2006.

For vaccine candidates against parasites like Leishmania, one of their most important features is whether they could effectively induce CD4+ and CD8+ T cell responses. However, in this manuscript, the authors only studied the antibody responses, limiting to their binding levels against crude antigens, while no analysis of T cell responses has been performed at all. This may also be the reason why there was no correlation observed between splenic parasite load and antibody production as shown in Figure 5. Therefore, the conclusion of the analysis may be too weak.

In fact, the lack of specific reagents to evaluate the immunological response in hamsters limit the analysis of cellular immune response in this experimental model. However, this experimental model is very important due to its great similarity with the disease development presented by dogs and humans. Moreover, as it is already known that the high production of anti-Leishmania antibodies is a characteristic of disease progression, it is not expected to have a positive correlation between antibodies and parasitic load in vaccinated and infected animals (control groups). In this sense, the lack of positive correlation between antibodies and parasitic load in LAPSmP and LAPSmG groups reaffirms the effectiveness of vaccines in reducing the parasite load and protecting against disease progression.

Reviewer 2 Report

Leishmania amazonensis is one of the most important species causing cutaneous leishmaniasis in Brazil. In this study, the association between two polymeric nanoformulations loading Leishmania amazonensis antigens was evaluated as a potential immunobiological agent against VL using golden hamsters as an experimental model. The work is meaningful.

Specific comments:

1. The language of the manuscript should be improved.

2. Nanovaccine must be supplemented with relevant test results, such as SEM, TEM, Particle size analysis and Vaccine proportion.

3. Figure 1 Incomplete information display, please improve relevant information.

4. Figures 2 and 3 lack relevant data analysis.

5. The liver and spleen were used to evaluate the vaccine efficacy by qPCR. How to prove the reliability of the method?

6. How to explain the relationship between serum antibody titer and vaccine effect?

Author Response

Review Report 2

Leishmania amazonensis is one of the most important species causing cutaneous leishmaniasis in Brazil. In this study, the association between two polymeric nanoformulations loading Leishmania amazonensis antigens was evaluated as a potential immunobiological agent against VL using golden hamsters as an experimental model. The work is meaningful.

Specific comments:

  1. The language of the manuscript should be improved.

The manuscript was revised by a native English speaker to improve the text quality.

  1. Nanovaccine must be supplemented with relevant test results, such as SEM, TEM, Particle size analysis and Vaccine proportion.

We have added further information (Table 1) about the submicrometric particle characterization, as requested, that was incorporated in the revised version of manuscript.

  1. Figure 1 Incomplete information display, please improve relevant information.

The clinical parameters are used in hamster model to assess the disease progression. In this sense, the Fig. 1 was included in the manuscript to demonstrate a distinct profile related to VL ongoing based on clinical feature: control groups (C, PSmG, PSmG, and LA) displaying a prominent dermatological change in contrast to groups with nanoparticles loaded with L. amazonensis antigens (LAPSmG and LAPSmP).

  1. Figures 2 and 3 lack relevant data analysis.

We have described the statistical analysis in Fig 2 and Fig 3. In Fig 2 the statistical differences (p<0.05) were represented by acronyms of the corresponding group (PI: Pre-immune; C: Control; SAP: Saponin; PSmP: Small Empty Submicron Particle; PSmG: Large Empty Submicron Particle; LA: L. amazonensis crude antigen; LASAP: L. amazonensis crude antigen associated with saponin; LAPSmP: Small Submicron Particle associated with L. amazonensis; LAPSmG: Large Submicron Particle associated with L. amazonensis). Similar strategy was used in Fig 3 that displayed the weights of live and spleen, important organs affected by L. infantum infection. The left panel of Fig 3 demonstrate the means and standard deviation of liver (A) and spleen (B) weights. Furthermore, in the right panne of Fig 3 we have included representative pictures of liver and spleen considering each analyzed group.

Importantly, similar strategy was explored by Moreira et al. (2012; DOI 10.1186/s13071-016-1464-y) describing clinical, hematological and biochemical alterations in hamster after L. infantum infection.

  1. The liver and spleen were used to evaluate the vaccine efficacy by qPCR. How to prove the reliability of the method?

The parasitism by L. infantum in hamsters presents high burden in liver and spleen as previously described (Moreira et al., 2012 - doi:10.1371/journal.pone.0047907). Moreover, it was demonstrated that wild L. infantum strain presenting high virulence profile displayed positive Spearman’s correlation between organ weight (liver or spleen) and parasite load evaluated by quantitative real-time PCR (Moreira et al., 2016 - DOI 10.1186/s13071-016-1464-y). These data support the application of qPCR in liver and spleen samples from L. infantum infected hamsters as reliable methodology. Furthermore, our group have been published an efficacy analysis of distinct vaccines against VL in hamster using qPCR supporting the application of qPCR in liver and spleen samples in this animal model (Ottino et al., 2022 - https://doi.org/10.3390/vaccines10111848).

  1. How to explain the relationship between serum antibody titer and vaccine effect?

The antibody production is an important feature in VL progression (REIS et al., 2006) and high antibody levels in addition to Leishmania antigens induce the formation of circulating immune complexes. These immune complexes are responsible for some of the clinical manifestations, including renal failure, the main clinical manifestation, which leads to the death of the infected animal (CACHEIRO‐LLAGUNO et al., 2021). In fact, several studies have already demonstrated a direct relationship between tissue parasite density, clinical status and antibody titres in canine VL (Reis et al., 2006; Dos-Santos et al., 2008; Leal et al., 2014; Laranjeira et al., 2014; Proverbio et al., 2014). As the hamster model presents high similarity to the progression of the disease in dogs, antibody production is also detected in hamster model that is associated with disease ongoing (Moreira et al., 2012). Concomitantly, clinical alterations are detected in these animals (Moreira et al., 2012). As it is known that VL progression is marked by a high production of anti-Leishmania antibodies (Reis et al., 2006), the positive correlation between the parasite load and antibody production detected only in Control and Sap groups reaffirms the vaccinal capacity of the vaccines tested in the present study.

Dos-Santos, W., Jesus, E., Paranhos-Silva, M., Pereira, A., Santos, J., Baleeiro, C., et al. (2008). Associations among immunological, parasitological and clinical parameters in canine visceral leishmaniasis: emaciation, spleen parasitism, specific antibodies and leishmanin skin test reaction. Vet. Immunol. Immunopathol. 5, 3–4. doi: 10.1016/j.vetimm.2008.02.004

de Almeida Leal, G., Roatt, B., de Oliveira Aguiar-Soares, R., Carneiro, C., Giunchetti, R., Teixeira-Carvalho, A., et al. (2014). Immunological profile of resistance and susceptibility in naturally infected dogs by Leishmania infantum. Vet. Parasitol. 205, 472–482. doi: 10.1016/j.vetpar.2014.08.022

Proverbio, D., Spada, E., Bagnagatti de Giorgi, G., Perego, R., and Valena, E. (2014). Relationship between Leishmania IFAT titer and clinicopathological manifestations (clinical score) in dogs. Biomed. Res. Int. 2014:412808. doi: 10.1155/2014/412808

Laranjeira DF, Matta VL, Tomokane TY, Marcondes M, Corbet CE, Laurenti MD. Serological and infection statuses of dogs from a visceral leishmaniasis-endemic area. Rev Saude Publica. 2014 Aug;48(4):563-71. doi: 10.1590/s0034-8910.2014048005224.

Moreira, D.; Vitoriano-souza, J.; Roatt, B. M.; Melo, P.; Vieira, D. A.; Ker, H. G.; Mirelle, J.; Cardoso, D. O.; Giunchetti, R. C.; Carneiro, M.; Lana, M.; Reis, A. B. Parasite Burden in Hamsters Infected with Two Different Strains of Leishmania (Leishmania) infantum: “Leishman Donovan Units” versus Real-Time PCR. PLoS One 2012, 7, 10, e47907, doi: 10.1371/journal.pone.0047907.

REIS, A. B. et al. Phenotypic features of circulating leucocytes as immunological markers for clinical status and bone marrow parasite density in dogs naturally infected by Leishmania chagasi. Clinical and Experimental Immunology, v. 146, n. 2, p. 303–311, 2006.

Round 2

Reviewer 1 Report

The authors addressed the revision comments in details, and added several sentences in corresponding sections to improve and complete the description. It may be better if the authors could add more content discussing the limitation of this study, especially regarding the T cell response as well as the animal model itself.

Author Response

As requested, we have included additional information inDiscussion Section, as folow:

"(...) In this sense, the limitation to evaluate the cellular immune response in addition to the production of cytokines are the main negative points of the hamsters animal model in pre-clinical vaccine trials. Furthermore, the VL ongoing is marked by a high production of anti-Leishmania antibodies and a positive correlation between antibody production and disease progression has already been reported [33, 51, 52]. The evaluation of parasite load is the main parameter to be evaluated for the analysis of efficacy, regardless of the animal model used in the pre-clinical test. Despite the limitation of evaluation in the cellular immune response in hamsters, this animal model is highly susceptible to L. infantum infection, which is a critical factor for the analysis of vaccine efficacy. (...)".

Reviewer 2 Report

Accept in present form

Author Response

We acknowledge the Reviewer#2 for the comments of our manuscript.